# Characterizing and Verifying Scientific Claims: Qualitative Causal Structure is All You Need

**Jinxuan Wu**♠    **Wenhan Chao**♠    **Xian Zhou**♣*    **Zhunchen Luo**♣*

♠School of Computer Science and Engineering, Beihang University,
♣Information Research Center of Military Science, PLA Academy of Military Science,
{wujinxuan, chaowenhan}@buaa.edu.cn
zhouxian@alumni.sjtu.edu.cn, zhunchenluo@gmail.com

## Abstract

A scientific claim typically begins with the formulation of a research question or hypothesis, which is a tentative statement or proposition about a phenomenon or relationship between variables. Within the realm of scientific claim verification, considerable research efforts have been dedicated to attention architectures and leveraging the text comprehension capabilities of Pre-trained Language Models (PLMs), yielding promising performances. However, these models overlook the causal structure information inherent in scientific claims, thereby failing to establish a comprehensive chain of causal inference. This paper delves into the exploration to highlight the crucial role of qualitative causal structure in characterizing and verifying scientific claims based on evidence. We organize the qualitative causal structure into a heterogeneous graph and propose a novel attention-based graph neural network model to facilitate causal reasoning across relevant causally-potent factors. Our experiments demonstrate that by solely utilizing the qualitative causal structure, the proposed model achieves comparable performance to PLM-based models. Furthermore, by incorporating semantic features, our model outperforms state-of-the-art approaches comprehensively.[1]

## 1 Introduction

A scientific claim refers to a statement or proposition that puts forward a particular hypothesis, theory, or discovery within the realm of science. The process of verifying a scientific claim typically entails thorough examination and assessment to ascertain its validity, relying on evidences that can support, refute, or remain neutral regarding the claim. In order to accomplish this, researchers must diligently scrutinize and characterize the specific features, qualities, or attributes of the claim. They

are tasked with identifying the essential elements, variables, or factors involved and furnishing a comprehensive comprehension of the claim's essence and extent.

In the field of scientific claim verification, the majority of research efforts have been devoted to dedicating attention architectures (Zhang et al., 2021) and leveraging the text comprehension capabilities of Pre-trained Language Models (PLMs) (Sarrouti et al., 2021; Wadden et al., 2022) to improve performance. These studies follow the end-to-end paradigm and employ transformers as the foundational framework, enabling the model to focus on relevant parts of the claim and capture the essential elements and relationships within the text. However, despite their proficiency to capture key elements and their interactions, these models neglected the causal structure information inherent in the claims that should be characterized during the process of scientific claim verification, let alone establish a comprehensive chain of causal inference between the claim and the evidence.

Following the schema proposed in SciClaim (Magnusson and Friedman, 2021), we provide an example of qualitative causal structure in a scientific claim in Figure 1. The claim describes the relationship between 'temperature' and 'COVID-19 transmission', which we identify as the key *factors* (highlighted in green). The core meaning conveyed in the claim is the causal connection between these factors, expressed by the term 'change' indicating an *association* (highlighted in red in Figure 1). The arrows from *association* to *factors* indicate that 'temperature' serves as the cause, while 'COVID-19 transmission' is effected by this cause. The yellow spans represent *modifiers* of the *association*, denoting its likelihood, extent, or direction, constraining the applicability or scope of the claim. As shown in Figure 1, there could also exist a qualitative causal structure in the evidence.

Unlike factoid claim verification, a cognitive sys-

---

* Corresponding authors.

[1]Our code is available at: https://github.com/VulnDetector/VerQCS.

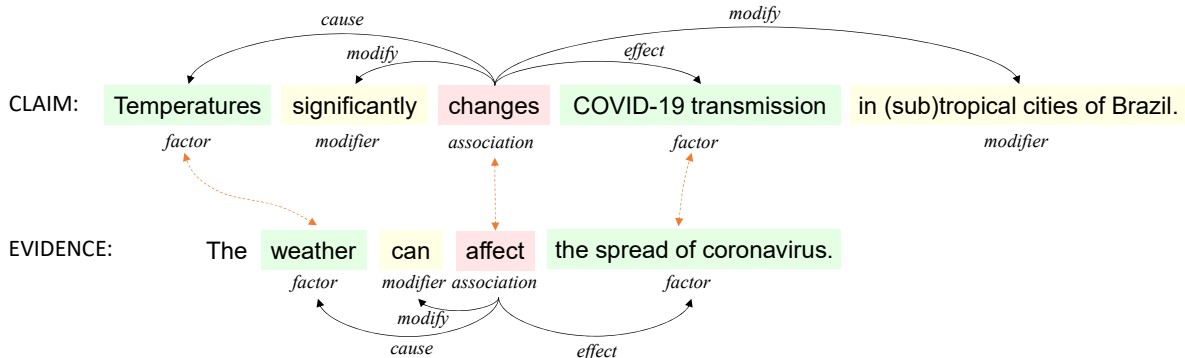

Figure 1: An example of qualitative causal structures present in a claim-evidence pair.

tem developed for verifying scientific claims is anticipated to make sense of specialized descriptive language and employ causal reasoning concerning interactions between quantities or entities (Wadden et al., 2020). Of utmost importance is the understanding of the qualitative causal structure, as we hypothesize that *it can characterize a scientific claim and facilitate the traversal of causal reasoning among relevant causally-potent factors for verification*. For instance, the evidence-based verification of the claim in Figure 1 can be achieved by exploring the relevant key *factors* and the *associations* that aggregate features from factors. Moreover, the *modifiers* play a crucial role in determining the direction of traversal, e.g., replacing 'significantly' with 'hardly' or 'never' could completely change the meaning of the claim.

With the hypothesis, the objective of this paper is to examine the role of qualitative causal structure to characterizing and verifying scientific claims. To achieve this goal, we employ the schema introduced in SciClaim (Magnusson and Friedman, 2021) to construct simplified qualitative causal structures for claim-evidence pairs in the datasets HEALTHVER (Sarrouti et al., 2021) and SCINLI (Sadat and Caragea, 2022). These qualitative causal structures are then organized as heterogeneous graphs to enable reasoning. Further, we propose a novel attention-based graph neural network model that performs causal reasoning between claim and evidences. Considering that compound claims need to be divided into several atomic subclaims, we introduce a graph decomposition algorithm and compute representations for sub-graphs, which are then aggregated using a weighted-sum operation to obtain final representations for the entire graph. The weight distribution

of sub-graphs also provides insights into which sub-claims are supported by evidence and which are not. Through extensive experiments, we demonstrate that our proposed model, utilizing only the qualitative causal structure information, achieves comparable performance to Pre-trained Language Model (PLM)-based models. Additionally, by incorporating the semantic features of the original sentences, our model achieves state-of-the-art performance.

## 2  Related Work

### 2.1  Scientific Claim Verification

The fact-checking task aims to design a model to automatically verify given claims. The FEVER task (Thorne et al., 2018) is a typical fact-checking dataset where claims are human-generated, and the evidence is extracted from Wikipedia. Recently, verifying scientific claims has drawn significant attention from the NLP community. There are serveral scientific claim verification datasets: Sci-Fact (Wadden et al., 2020), PubHealth (Kotonya and Toni, 2020), COVID-Fact (Saakyan et al., 2021), HealthVer (Sarrouti et al., 2021). Especially, HealthVer constructs claims from snippets returned by a search engine for questions about COVID-19. According to (Sarrouti et al., 2021), the claims from HealthVer are more complex, which is a more real-world situation setting.

Most current research formulates the claim verification task as performing the Natural Language Inference (NLI) task (Chen et al., 2017; Ghaeini et al., 2018; Parikh et al., 2016; Radford et al., 2018; Peters et al., 2018; Luken et al., 2018). Similarly, in scientific claim verification, most existing work concatenates the claim and the evidence together to establish the semantic relation between

them. ARSJoint (Zhang et al., 2021) uses a hierarchical attention network (Pappas and Popescu-Belis, 2017) to perform word attention and sentence attention. MultiVerS (Wadden et al., 2022) concatenates the claim and each sentence of evidence together. By leveraging the NLI ability of the pre-trained language model and the pre-training procedure, MultiVerS achieved state-of-the-art performance on the SciFact dataset. Similarly, the best performing model (Sarrouti et al., 2021) on the HealthVer dataset also leverages the NLI ability of the pre-trained model T5.

## 2.2 Causal structure in claims

Several recent studies have integrated structural reasoning process into their models. GEAR (Zhou et al., 2019) proposes that interaction among sentences in the evidence should be considered and constructs a fully-connected evidence graph by concatenating the claim with each sentence in the evidence as a node and connecting all node pairs with edges. Similarly, KGAT (Liu et al., 2020) builds a graph in the same way but provides more fine-grained evidence selection using neural matching kernels. In cognative system, the schema proposed by SciClaim (Magnusson and Friedman, 2021) provides a meaningful approach. They develop multiple entity types such as *Factors*, *Association*, and *Magnitudes*, as well as construct different relations between these entities. The schema is adopted to simplify complex claims (Wührl and Klinger, 2022) to solve the mismatch between trained models on SciFact and real-world claims.

## 2.3 Graph Representation and Reasoning

We also refer to the work of heterogeneous graphs when designing the model. Different from homogenous graphs, heterogeneous graphs (Sun and Han, 2013) contain multiple node or relation types, posing unique challenges for graph-based learning methods like Graph Convolutional Network (GCN) and Graph Attention Network (GAT) that can only handle homogenous graphs. To address this issue, meta-path (Sun et al., 2011) is a typical method that leverages rich information in heterogeneous graphs, but it requires human knowledge to define meta-paths. HERec (Shi et al., 2018) is a typical meta-path-based work that extracts several homogenous graphs using meta-path and performs aggregation on the representations learned on the homogenous graph respectively. Additionally, there has been research on applying heterogeneous graphs to NLP

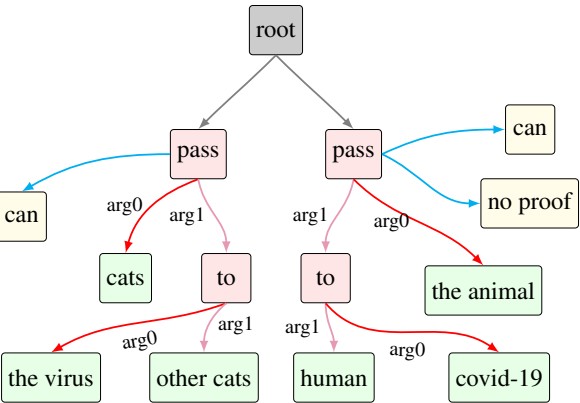

*Cats can pass the virus to other cats, they discovered no proof that the animals can pass COVID-19 to humans.*

Figure 2: An illustrative graph depicting the qualitative causal structure within the scientific claim. The edges in blue are *modifying* relations.

(Hu et al., 2020; Linmei et al., 2019; Hu et al., 2020) since graph is a powerful tool to represent entities and their complex interactions.

## 3 Qualitative Causal Schema

SciClaim (Magnusson and Friedman, 2021) proposes a fine-grained graph schema for scientific claims. We simplified the schema by removing certain entity and relation types that are not essential for the task as follows.

*Entities* are labeled text spans. We aggregate six entity types into three: **factor**, **association**, and **modifier**, furthermore, to conduct aggregation, we design a novel node type **root**. **Factor** remains unchanged as variables tested or asserted within a claim, typically nouns or noun-phrases, such as 'cats' and 'covid-19' in Figure 2. **Association** are the same in the original schema, which is explicit phrases associating one or more factors that describe the structural effect of the entity, such as 'pass' in Figure 2. **Modifier** is a new entity type, which aggregates the original entity types *magnitude*, *evidence*, *epistemics*, and *qualifiers* since they perform similarly in scientific claim verification as modifiers of the *association*. **Root** is designed to aggregate tree representations into a single vector representing the whole scientific sentence.

*Relations* are directed edges between labeled entities, and we preserve the original **arg0** and **arg1** relation types, which relate *associations* to their cause or effect, respectively. We also retain the modifier edge with a new name **modifying** to distinguish the entity *modifier*. *Modifying* relates associations to their modifying components (*modifiers*

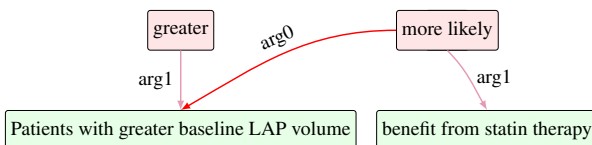
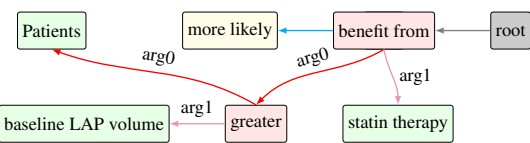
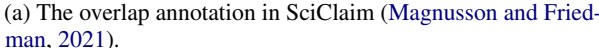

(a) The overlap annotation in SciClaim (Magnusson and Friedman, 2021).

(b) Our proposed non-overlapped nested tree design annotation.

Figure 3: The entities in red, yellow and green are *associations*, *modifiers* and *factors* repectively. We illustrate a typical entity-overlapped annotation modification with the text: *Patients with greater baseline LAP volume are more likely to benefit from statin therapy.*

in our schema), and **comp_to** which represents comparison in scientific claim verification. However, we remove other high-order edges such as $q+$ and $q-$, since these relationships are contained in the semantics of *associations*.

In addition to modifying the schema, we propose the introduction of a nested tree structure to replace the annotation setting that allows for entity overlapping. The original schema permits the overlapping of two entity spans, posing a challenge for models to accurately identify and differentiate between distinct entities. We present a typical example of entity overlapping in Figure 3a, where the entire span 'Patients with greater baseline LAP volume' is annotated as a *factor*, despite the overlapping *association* 'greater'. Furthermore, the schema does not enable the subsequent model to concentrate on specific scientific entities within the sentence, such as 'baseline LAP volume' and 'statin therapy', as these concepts are not extracted as entities. Therefore, we propose the nested design to decompose the overlapped entity into smaller, non-overlapping sub-trees as we present in Figure 3b, the span 'Patients with greater baseline LAP volume' in Figure 3a is decomposed into a sub-tree rooted by 'greater' in Figure 3b. The tree structure enhances the model's comprehension of hierarchical relationships between entities and their modifiers, thereby facilitating more precise identification of relevant entities and relations for claim verification.

Following the schema, we annotate the structure of each sentence in HEALTHVER dataset(Sarrouti et al., 2021), which is a complex scientific claim verification dataset. We proceed to train a joint entity and relation extraction model for extracting the sentence structures. Additionally, we utilize the extraction model to conduct experiments on another dataset SciNLI (Sadat and Caragea, 2022).

# 4 Proposed Approach

In this section, we introduce the proposed model for scientific claim **Ver**ification by utilizing the **Q**ualitative **C**ausal **S**tructure, also referred to as **VerQCS**. We explain the big picture of VerQCS here. VerQCS utilizes causal graphs within claims and evidence. The graph structures are extracted by our dedicated extraction model. With both the graphs and the textual content, VerQCS performs reasoning and ultimately predicts the relationship. Modules like factor cross-attention, causal graph decomposition, and causal graph reasoning are introduced to capture the structural characteristics of claims and evidence, which are subsequently employed to predict the relationships.

Prior to introducing the model, two properties of the graph structure are introduced:

First, the *associations* always serve as the *root* nodes of the trees. In Figure 2, both sub-claims have 'pass' as their root, which is an *association*;

Second, if we designate the tree directly connected to the *root* as the root tree, we observe that root tree corresponds to the semantic span of sub-claims or sub-evidence. In Figure 3b, the sub-tree rooted by 'greater' corresponds to the noun-property phrase 'patients with greater baseline LAP volume' which does not provide any assertive information that can be argued or serve as evidence. We have observed that only the root tree corresponds to the semantic span, which is the tree rooted by 'benefit from' in the figure.

The properties we observe reveal the intrinsic connections between structure representations and semantic meaning. Sub-claim or sub-evidence can be extracted via identifying the root tree. Consequently, we employ structural reasoning on their respective root trees to facilitate reasoning between sub-claims and sub-evidence.

The tree representation learning module, which we further detail in section 4.5, encodes a tree structure using an embedding vector and employs a re-

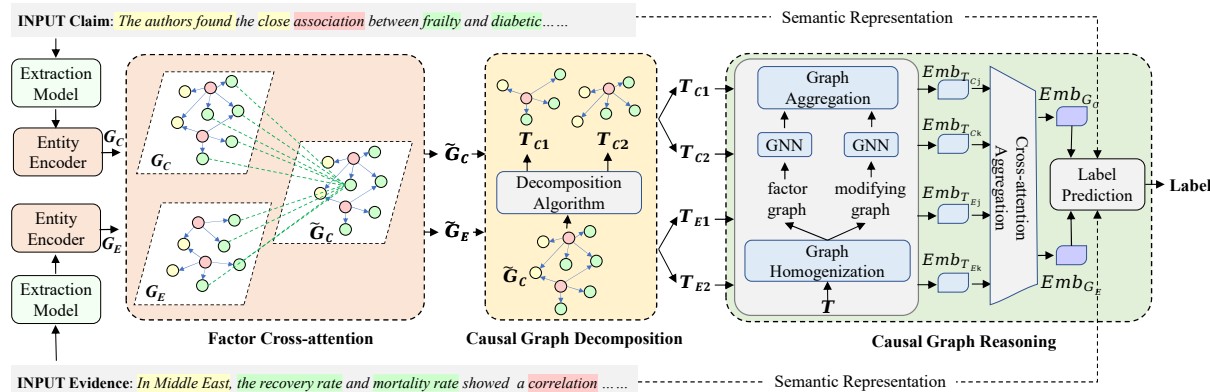

Figure 4: The framework of VerQCS with an input example of a claim-evidence pair. The nodes in red, yellow and green represent *associations*, *modifiers* and *factors* respectively.

cursive design to handle the nested tree structure. For instance, Figure 3b illustrates a sample containing two trees. The input of the tree representation module is the root tree rooted by 'benefit from'. Upon parsing the tree structure, the module identifies a dependent sub-tree rooted by 'greater'. Subsequently, the module recursively calls itself, generating an embedding of the sub-tree that is subsequently utilized to generate the embedding of the root tree.

## 4.1 Joint Entity and Relation Extraction

Using the annotated structure representations of sentences in HEALTHVER, we train a joint model to extract the structure representations. VerQCS utilizes the predicted structures as input to perform structural reasoning and predicts the final relationships within the claim-evidence pairs.

Illustrated in Figure 4, the extraction model takes the claim and evidence texts as input. By representing entities as nodes and relationships as edges, the extraction model predicts the structural graph of the claim ($G_C$) and evidence ($G_E$).

## 4.2 Entity Encoder

The entity encoder generates embeddings for the extracted entities. The module consists of a token encoder and an entity pooling. Considering the scientific property, we use SciBERT (Beltagy et al., 2019) as the token encoder to generate token embeddings. Let $entity_i = (h_{i_1}, h_{i_2}, \dots, h_{in_i})$ denote the embeddings of the tokens in $entity_i$, we employ an attention-based entity representation to perform entity pooling:

$$\alpha_{i,k} = \frac{exp(W \cdot h_{ik} + b)}{\sum_{j=i_1}^{in_i} exp(W \cdot h_{ij} + b)}$$
$$e_i = \sum_{k=1}^{n_i} \alpha_{i,k} \cdot h_{ik} \quad (1)$$

Where $e_i$ is the entity representation of $entity_i$, the matrix $W$ maps the hidden representation of token $h$ into a scaler.

## 4.3 Factor Cross-Attention

We perform factor cross-attention between $G_C$ and $G_E$, initializing each node with the corresponding entity embedding. The motivation of the component is that we hypothesize the interaction between *factors* is crucial for subsequent reasoning. This interaction should not only occur within the scientific claim or evidence individually, but also between the pair. Intuitively, if the evidence reasons the claim, it is likely that the *factors* share commonalities, such as co-reference or domain-specific relationships. We concatenate all the factor entity representations: $F = [f_{c_1}, \dots, f_{c_{n_c}}, \dots, f_{e_1}, \dots, f_{e_{n_e}}]$, where $n_c$ and $n_e$ are the numbers of *factors* in claims and evidence respectively, therefore $F \in \mathbb{R}^{(n_c+n_e) \times d_h}$, where $d_h$ is the dimension of the entity representation. To perform *factor* interactions, we employ cross-attention on *factors* from both sides:

$$F_{new} = (FV) \cdot softmax(\frac{(FQ) \cdot (FK)^{\mathrm{T}}}{\sqrt{d_k}}) \quad (2)$$

$Q$,$K$,$V$ are the matrixes which map $d_h$-dimension vectors into $d_q$,$d_k$ and $d_v$ dimensional vector space respectively. All *factor* in $G_C$ and $G_E$ are then updated with the new embeddings, leading to the new graphs $\widetilde{G}_C$ and $\widetilde{G}_E$.

## 4.4 Causal Graph Decomposition

Since the graph forms the structure of nested trees with root trees containing specific semantic meanings, we design the module to decompose the causal graph into trees and identify root trees. In Figure 4, since there are two trees in $\widetilde{G}_C$, $T_{C1}$ and $T_{C2}$ are the decomposition results. The pseudocode is presented in Algorithm 1. The output $E$ and $R$

---

**Algorithm 1:** Decomposition Algorithm

**Input:** The graph of the scientific sentence:
  Entities: $\mathcal{E} = \{e_1, e_2, \ldots, e_n\}$;
  Relations: $\mathcal{R} = \{r_{ij} | e_i \xrightarrow{r_{ij}} e_j\}$.

**Function** TreeGrow$(e, E, R, flags)$:
  **foreach** *entity in* $\{e_i | e \xrightarrow{rel} e_i\}$ **do**
    $E = E \cup entity$;
    $R = R \cup rel$;
    **if** *type(entity) = association* **then**
      $flags_{id(entity)}$ = False;
      **return** $E, R, flags$;
    **else**
      TreeGrow$(entity, E, R, flags)$;
  **return** $E, R, flags$;

associations set: $\mathcal{A} = \{a_1, a_2, \ldots, a_m\} \subseteq \mathcal{E}$;
$flags = list(m)$;
Initialize $flags$ with True;
**foreach** *r in* $\mathcal{A}$ **do**
  $E = \phi$; $R = \phi$;
  $E, R, flags$=TreeGrow$(r, E, R, flags)$;

**Output:** $E, R, flags$.

---

represent the entities and relations of each tree, respectively, with $r$ as the root. The $flags$ variable indicates root trees.

## 4.5 Causal Graph Reasoning

**Tree Representation Learning**   The objective of the proposed component is to represent the tree structure using embeddings, thereby preserving the structural information. However, since the tree is a heterogeneous graph with multiple relations and entities, we propose graph homogenization as a solution to split the heterogeneous graph into two graphs: the **factor graph** and the **modifying graph**. The factor graph is obtained by preserving the *factors* and *associations* with their connecting edges, while the modifying graph consists of the *associations* and *modifiers* with *modifying* relations. This division is based on the notion that the *factors* are noun properties, while the *modifiers* modify the di-

rection or credibility of the *association*. These two graphs offer distinct perspectives and contribute to a comprehensive understanding.

Graph neural networks are employed to generate embeddings for these graphs. The embeddings of the individual graphs are learned independently and subsequently concatenated to form the final embedding of the tree.

$$Emb_T = W_{tree}(tree_{factor} || tree_{modifying}) \quad (3)$$

**Cross-attention Aggregation**   The module combines the embeddings of individual trees to create the graph embedding. For example, in Figure 4, the embedding of $\widetilde{G}_C$, denoted as $Emb_{G_C}$, is calculated based on the embeddings of $T_{C1}$ and $T_{C2}$. Since sub-claims and sub-evidence are represented by root trees, the aggregation process focuses on the embeddings of these root trees.

The rationale behind the aggregation is that if a sub-evidence tree strongly reasons a sub-claim tree, the weight of the sub-evidence should be assigned a higher value when calculating the embedding of *root* for evidence, and the same applies to the claim. To describe the relationships between trees in the scientific evidence and claim, we utilize a matrix $S \in \mathbb{R}^{n_c \times n_e}$, where $n_c$ and $n_e$ denote the number of root trees in the claim and evidence, respectively. The aggregation process is conducted by leveraging the matrix $S$ as follows:

$$s_{ij} = \frac{W_{sim}Emb_{T_{Ci}} \cdot W_{sim}Emb_{T_{Ej}}}{\sqrt{d_{sim}}}$$
$$w_{claim} = softmax(\max_{dim=0} S)$$
$$root_{claim} = \sum_{i=1}^{n_c} w_{claim_i} \cdot Emb_{T_{Ci}} \quad (4)$$
$$w_{evi} = softmax(\max_{dim=1} S)$$
$$root_{evi} = \sum_{j=1}^{n_e} w_{evi_j} \cdot Emb_{T_{Ej}}$$

Let $s_{ij}$ denote the scalar value in matrix $S$, $W_{sim}$ be the mapping function that projects the tree embedding into a $d_{sim}$-dimensional space, and $w_{claim}$ and $w_{evi}$ represent the computed weights for claim and evidence root trees, respectively. Additionally, $\frac{1}{\sqrt{d_{sim}}}$ serves as the scaling factor, $\max_{dim=0}$ generates an $n_c$-dimensional vector by selecting the maximum value along each row, and $\max_{dim=1}$ indicates selection along each column.

**Label Prediction** In our final prediction, we acknowledge the value of semantics as an advantageous complement to the structural representation. Consequently, we concatenate the claim and evidence tokens using `[SEP]` as a separator, considering the output of the `[CLS]` token as the semantic feature. Subsequently, we combine the structural and semantic representations of the pair, employing cross-entropy as the loss function.

## 5 Experiment

In this study, we perform two distinct experiments: structure extraction and scientific claim verification, respectively. We present the experiment of scientific claim verification in the section, more details about structure extraction are presented in Appendix A.1.

### 5.1 Experimental Setup

**Dataset** The extraction model is trained on the training set of HEALTHVER since we annotate the structure on the dataset. We train VerQCS on the training set with the ground truth of the structures, and VerQCS is required to predict the cases in test set with the structure predicted by the extraction model. To evaluate how the error in the extraction model may propagate to VerQCS, we also leverage the ground truth structures in test for comparison. We name the experiment settings as HEALTHVER$_{pred}$ and HEALTHVER$_{truth}$ respectively. We also experiment on another dataset SciNLI (Sadat and Caragea, 2022) without any annotation. The structures in SciNLI are derived from the parsing model trained on HEALTHVER.

**Implementation Details** In scientific claim verification, We optimize VerQCS with AdamW (Loshchilov and Hutter, 2017) for 50 epochs with a learning rate of 4e-5 and weight decay of 1e-2.Considering the scientific nature of the task, we employ SciBERT (Beltagy et al., 2019) as the token embedding layer. Within the factor cross-attention module, we set the multi-head parameter to 3 to account for the three different roles. The entity dimension remains consistent at 768 for both factor cross-attention and tree representation learning. In the training process, we set the batch size to 1 with the accumulation step set to 32.

### 5.2 Overall Performance

We present the result of scientific claim verification experiment here. The performance of the extraction model is presented in Appendix A.1.

In our experiments conducted on the HEALTHVER dataset, we compare the performance of VerQCS with T5 (Raffel et al., 2020), which is the current state-of-the-art model, as well as similar PLMs such as BERT (Devlin et al., 2019) and SciBERT (Beltagy et al., 2019), as presented in Table 1. Since semantic models do not require structure input, there is no performance difference between the HEALTHVER$_{truth}$ and HEALTHVER$_{pred}$ settings.

Models denoted by StrOnly represent semantic-feature-free reasoning models, and the subscript indicates the adopted GNN method. In HEALTHVER$_{truth}$, the performance of StrOnly$_{GCN}$ is superior to BERT, when introducing the error in the parsing model, StrOnly models perform slightly worse than PLMs. Therefore, we hypothesize that by incorporating the extracted reasoning element information alone, the model can achieve comparable performance to models fine-tuned on PLMs. Furthermore, it is observed that when the StrOnly models are augmented with semantic features, VerQCS outperforms these PLMs, even in the presence of errors introduced by the parsing model.

We observe that in StrOnly models, GCN is a better choice than GAT, which appears to contradict findings in previous studies. We hypothesize that the input to the GNN consists of a decomposed factor or modifying graph on the extracted trees. These graphs consist of one *association* and several nodes with the same label, lacking complex structures and numerous nodes. Consequently, GAT, which has more parameters than GCN, may tend to overfit the input. Additionally, the attention layer in GAT may provide limited performance gains due to the presence of nodes belonging to the same class in the graph.

Additionally, we evaluate performance by comparing it with a large language model: GPT-4. To perform reasoning with GPT-4, we utilize the API provided by OpenAI. The prompts are designed with three distinct settings. "Instruct GPT-4" prompts contain solely the task description. "3-shot GPT-4" refers to demonstrations of GPT-4 using three sample cases, with one case for each label. "6-shot GPT-4" involves six cases, each featuring the task description, and two cases for each label. The results are represented in Table 2. We conclude that in the "Instruct GPT-4" setting, where only

| Model | HEALTHVER$_{truth}$ | | | HEALTHVER$_{pred}$ | | |
|---|---|---|---|---|---|---|
| | prec | rec | F1 | prec | rec | F1 |
| BERT | 75.19 | 73.71 | 74.17 | 75.19 | 73.71 | 74.17 |
| SciBERT | 80.51 | 78.14 | 78.71 | 80.51 | 78.14 | 78.71 |
| T5 | 81.10 | 80.23 | 80.52 | 81.10 | 80.23 | 80.52 |
| StrOnly$_{GCN}$ | 76.58 | 75.17 | 75.24 | 73.12 | 73.00 | 73.05 |
| StrOnly$_{GAT}$ | 74.89 | 73.66 | 74.10 | 73.01 | 72.34 | 72.66 |
| VerQCS | **82.79** | **81.36** | **82.14** | **81.10** | **81.65** | **81.33** |
| w/o Factor Cross-attention | 79.08 | 77.20 | 77.68 | 76.37 | 77.10 | 76.66 |
| w/o Causal Graph Decomposition | 81.10 | 80.23 | 80.52 | 79.21 | 79.93 | 79.45 |
| w/o Graph Homogenization | 76.67 | 75.22 | 75.64 | 74.90 | 75.32 | 75.11 |

Table 1: Experimental results of the comparison methods. StrOnly models exclusively utilize the qualitative causal structure information. And the models started with 'w/o' indicate the approaches considered in the ablation study.

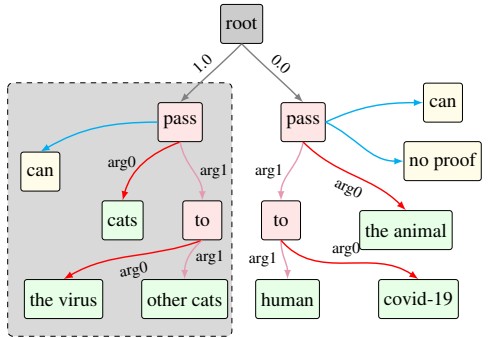

*Cats can pass the virus to other cats*, they discovered no proof that the animals can pass COVID-19 to humans.

(a) The weight distribution of the claim.

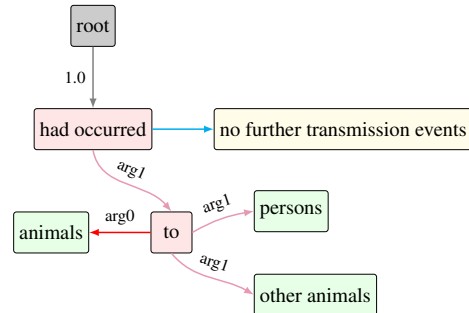

*No further transmission events from animals to other animals or persons had occurred.*

(b) The weight distribution of the evidence.

Figure 5: An example where the evidence *refutes* the claim.

| Prompt construction | F1 |
|---|---|
| Instruct GPT-4 | 68.08 |
| 3-shot GPT-4 | 68.50 |
| 6-shot GPT-4 | 71.05 |

Table 2: Performance of GPT-4 with different settings.

| Model | C | R | E | N | F1 |
|---|---|---|---|---|---|
| BERT | 77.46 | 71.74 | 75.09 | 76.47 | 75.19 |
| SciBERT | 80.30 | 74.18 | 75.90 | 79.76 | 77.53 |
| RoBERTa | 81.18 | 74.22 | **77.99** | 78.86 | 78.06 |
| XLNet | 81.53 | **75.95** | 77.63 | 77.63 | 78.18 |
| StrOnly$_{GCN}$ | 71.15 | 68.33 | 69.14 | 71.00 | 70.66 |
| StrOnly$_{GAT}$ | 72.30 | 67.77 | 70.17 | 62.42 | 68.16 |
| VerQCS | **83.46** | 75.20 | 77.44 | **80.08** | **79.27** |

Table 3: Experimental results on SciNLI.

the task instruction was provided, GPT-4's performance was less impressive. By providing GPT-4 with example cases and leveraging its in-context learning ability, performance improved as more examples were given. We observed that GPT-4's performance was comparable to fine-tuned BERT models. This aligns with findings from similar research (Liu et al., 2023).

Furthermore, we conducted experiments on SCINLI, and the corresponding performance is presented in Table 3. The StrOnly models exhibit lower performance compared to PLMs. We hypothesize that this discrepancy arises due to the pars-

ing model being trained on annotated data from HEALTHVER, which leads to difficulties in generalization to SCINLI. Nevertheless, VerQCS continues to outperform the PLMs. More details about the analysis are presented in Appendix A.1.

## 5.3 Ablation Study

To demonstrate the effectiveness of the designed components in our model, we conducted ablation

studies where we removed factor cross-attention, causal graph decomposition, and graph homogenization. Based on the results presented in Table 1, the ablation study provides evidence of the effectiveness of these components. Without factor cross-attention, the performance drops, indicating the interaction among *facotors* is the key to conduct verification. We remove the graph decomposition with model reasoning directly on the entire graph. The result demonstrates the effectiveness of the component. In our reasoning process on the heterogeneous graph, we partition the graph into a factor graph and a modifying graph. By conducting an ablation experiment that removes graph homogenization, we provide evidence of the method's effectiveness.

### 5.4 Case Study

In the case study, we aim to present the weight distribution assigned to the root trees by VerQCS with a typical case in Figure 5. As depicted in Figure 5a, the claim consists of two sub-claims, each of which is mapped to a root tree in the graph representation while the structure of the evidence forms a single tree in Figure 5b. The evidence refutes the claim since the first sub-claim conveys the conflicting information, although the second sub-claim is consistent with the evidence. VerQCS has the capability to reveal which sub-claim is reasoned by evidence by assigning the weight distribution over trees, the tree corresponds to the first sub-claim is assigned with weight 1.0 in the case. Therefore, the weight distribution provides insights into how the reasoning process is conducted in VerQCS, which can be leveraged to further help researchers.

We also illustrate a more complex case where both claim and evidence contain multiple root trees in Appendix A.2.

### 6 Conclusions

This paper introduces the utilization of qualitative causal structure in scientific claim verification. To this end, we propose the novel model VerQCS, which performs structural reasoning on the causal graph. The framework has been demonstrated to be effective even without semantic features enhancing. Additionally, VerQCS demonstrates its capability of selecting sub-claims and sub-evidence by assigning weights to the sub-graphs, which provides insights of how VerQCS conducts reasoning. In conclusion, we have substantiated the efficacy of

qualitative causal structures in claim verification. Moreover, we hypothesize that the factor entities within causal graphs can be seamlessly linked to external knowledge graphs, thus offering opportunities for further exploration.

### Limitations

We suppose the VerQCS contains two major limitations. The first limitation is the choice of the strcture extraction model, we leverage an entity relation extraction model SpERT, while the annotated structure adheres to the tree structure that closely resembles the dependency parsing task, we have not yet explored whether utilizing successful methods from dependency parsing could yield improved performance. The second limitation pertains to our method operating as a pipeline paradigm, where an extraction model predicts structures and VerQCS conducts further reasoning using the predictions as input. The experiment results reveal an issue of error propagation. we hypothesize that employing a joint model could be a more effective approach, and this avenue will be explored in our future work.

### Ethics Statement

Scientific claim verification and fact-checking tasks are widely studied tasks in NLP, and we do not identify any notable ethical concerns associated with these tasks. Furthermore, the datasets utilized in our study are publicly available. While the SciClaim schema of the graph is derived from a published work, we obtained permission and approval from the original author to modify the schema for better alignment with our task.

### Acknowledgements

This work was supported by National Natural Science Foundation of China (No. 61976221). We would like to express our gratitude to the authors of SciClaim for their invaluable assistance. We also extend our appreciation to the reviewers for their patience and valuable feedback.

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

| Statistics | *claim* | | | *evidence* | | |
|---|---|---|---|---|---|---|
| | **Train** | **Dev** | **Test** | **Train** | **Dev** | **Test** |
| entity/token | 3.13 | 3.15 | 3.32 | 3.99 | 4.67 | 3.91 |
| relation/entity | 1.25 | 1.22 | 1.45 | 1.21 | 1.19 | 0.88 |

(a) The statistics of HEALTHVER

| Statistics | *sentence*1 | | | *sentence*2 | | |
|---|---|---|---|---|---|---|
| | **Train** | **Dev** | **Test** | **Train** | **Dev** | **Test** |
| entity/token | 5.05 | 4.98 | 4.88 | 5.02 | 4.86 | 4.49 |
| relation/entity | 2.25 | 2.16 | 2.09 | 2.30 | 2.24 | 2.22 |

(b) The statistics of SCINLI

Table 4: The statistics of the datasets. The *entity/token* is the average number of entities divided by the average number of tokens, indicating the density of entities, *relation/entity* indicates the density of relations.

## A Appendix

### A.1 Additional Expriment Result

More details about the experiments are presented in the section.

**Joint Entity Relation Extraction** The extraction model is trained on the training set of HEALTHVER since we annotate the structure on the dataset. For the structure extraction task, we leverage SpERT(Eberts and Ulges, 2019) to perform the structure extraction, the batch size is set to 32 with 50 epochs. The performance of the model in structure extraction is presented in Table 5. The model can achieve an F1-score of over 80 in entity extraction. However, it struggles to perform well in relation construction, as this task requires strong logical reasoning ability.

| Label | Prec | Rec | F1-score |
|---|---|---|---|
| association | 91.05 | 90.00 | 90.52 |
| factor | 87.45 | 86.20 | 86.82 |
| modifiers | 82.02 | 85.98 | 83.95 |
| arg0 | 80.77 | 80.77 | 80.77 |
| arg1 | 80.19 | 73.07 | 76.46 |
| comp_to | 53.33 | 42.11 | 47.06 |
| modifying | 69.10 | 75.93 | 72.36 |

Table 5: the performance of the joint extraction model.

**Dataset Statistics** Additional dataset statistics are provided in Table 4, where both the entity/token ratio and the relation/entity ratio in SCINLI significantly surpass those in HEALTHVER. This discrepancy suggests that the extraction model inade-

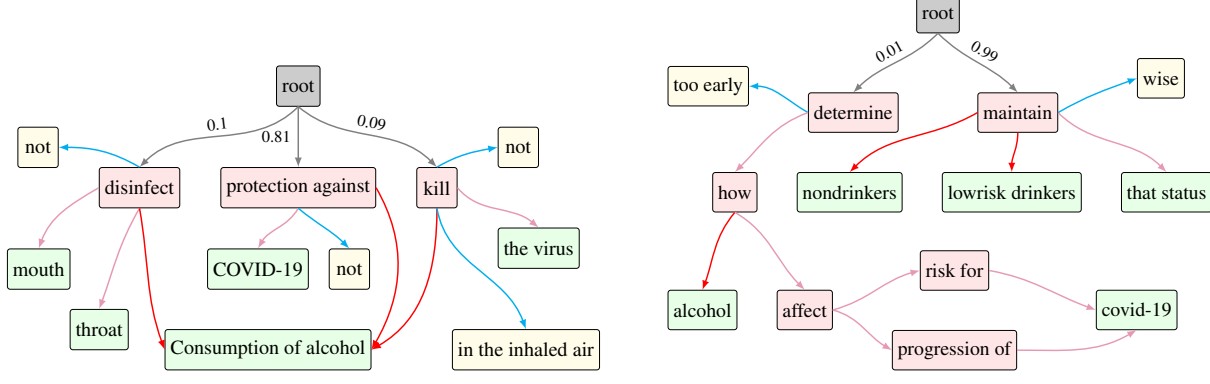

(a) The weight distribution of the claim.

(b) The weight distribution of the evidence.

**claims**: *Consumption of alcohol will not kill the virus in the inhaled air; it will not disinfect your mouth and throat; and it will not give you any kind of protection against COVID-19.*

**evidence**: *While it remains much too early to determine precisely how alcohol use might affect risk for or progression of COVID19, any efforts to connect the dots from past alcohol research would suggest it wise for nondrinkers and lowrisk drinkers to maintain that status during this public health crisis.*

Figure 6: a complex case where evidence *Supports* the claim.

quately extracts fine-grained reasoning elements in SCINLI.

## A.2 Complex Case

We illustrate a more complex case where both claim and evidence contain multiple trees in Figure 6, the weight distributions on trees of claim and evidence disclose the fine-grained relationships on tree level, which reveals the functional sub-claim and sub-evidence in the reasoning process.