# OpenReview forum: "Characterizing and Verifying Scientific Claims: Qualitative Causal Structure is All You Need"
_EMNLP/2023/Conference — EMNLP 2023 Main_

### Official Review · Reviewer_54T1 · 2023-08-04

**Soundness:** 3

**Excitement:**

4: Strong: This paper deepens the understanding of some phenomenon or lowers the barriers to an existing research direction.

**Paper Topic And Main Contributions:**

This paper proposes a graph-based method for verifying scientific claims. The authors introduce a complex pipeline made of 5 modules.  They present an attention-based graph neural network model that facilitates the reasoning of causally crucial factors in scientific claims.

**Reasons To Accept:**

- Experiments seem correctly conducted
- The model offers a means to decompose complex scientific claims into atomic subclaims, allowing for more detailed scrutiny and interpretation of the claims.

**Reasons To Reject:**

- The paper claims to compare with state-of-the-art models while the systems compared (BERT, SciBERT and T5) are quite old. Instruct LLMs are not compared.

**Reproducibility:**

4: Could mostly reproduce the results, but there may be some variation because of sample variance or minor variations in their interpretation of the protocol or method.

**Reviewer Confidence:**

2: Willing to defend my evaluation, but it is fairly likely that I missed some details, didn't understand some central points, or can't be sure about the novelty of the work.

---

> ### Author Rebuttal · Authors · 2023-08-29
>
> Thank you for reviewing our paper and raising your concerns about the work. Below, we'd like to provide our insights.
>
> **Q1**: The authors introduced a very complex pipeline and fail to explain clearly the big picture. I found the paper quite difficult to follow.
> **A1**: We appreciate your feedback and would like to provide a more concise response.
> - This paper focuses on emphasizing the pivotal role of qualitative causal structures in characterizing and verifying scientific claims based on evidence. We organize these structures into a heterogeneous graph and propose an attention-based graph neural network called VerQCS to facilitate causal reasoning across relevant causally-potent factors. Our pipeline comprises three main steps: 1) causal structure extraction (Section 4.1), 2) entity/factor representation learning (Sections 4.2 and 4.3), and 3) causal graph decomposition and reasoning (Sections 4.4 and 4.5).
> - We regret any potential confusion by not including an introduction to the general framework in Section 4, as there is so much we would like to present but space does not permit. We deeply regret any inconvenience this may have caused. In response, we are committed to a careful paper revision, which will include an introduction to the overall framework and a more comprehensive explanation of each model component's motivation.
>
> **Q2**: The examples justifying each component are not convincing enough.
> **A2**: We appreciate your feedback. To clarify the motivation behind the examples, we provide the following explanations:
> - Figure 1 introduces the use of causal structures for verification.
> - Figure 2 illustrates the schema of the heterogeneous graph.
> - Figure 3 highlights the differences between the original and modified schemas through a typical example.
> - Figure 5 presents a graph representation of a case study, showcasing the enhanced capabilities of VerQCS, which we believe can contribute to the understanding of performance improvements.
> We hope these clarifications provide a more compelling rationale for the examples used in our paper.
>
> **A3**: The paper claims to compare with state-of-the-art models while the systems compared (BERT, SciBERT and T5) are quite old. Instruct LLMs are not compared.
> **Q3**: Thanks for your question. Your insights have made a meaningful contribution to our future endeavors. We were also curious about the performance of Large Language Models (LLMs) in the reasoning task, so we conducted experiments using the GPT-4 API. The results are as follows:
> | Approach                     | F1 score   |
> | ---------------------------------- | ----- |
> | Instruct GPT-4                   | 68.08 |
> | 3-shot GPT-4                 | 68.50 |
> | 6-shot GPT-4                 | 71.05 |
> | BERT                                   | 74.17 |
> | $StrOnly_{GCN}$                  | 75.24 |
> | $StrOnly_{GAT}$                  | 74.10 |
> | $VerQCS$                         | 82.14 |
>
> We evaluated GPT-4 in different scenarios:
> - In the "Instruct GPT-4" setting, where only the task instruction was provided, GPT-4's performance was less impressive.
> - By providing GPT-4 with example cases and leveraging its in-context learning ability, performance improved as more examples were given. We observed that GPT-4's performance was comparable to fine-tuned BERT models. This aligns with findings from a related paper[1].
>
> In summary, our results suggest that while Large Language Models (LLMs) demonstrate robust semantic understanding of general text, their capabilities in reasoning for scientific claim verification tasks are an area worthy of further exploration. Your attention to this matter is greatly appreciated.
>
> Once again, we extend our gratitude for your meticulous assessment of our paper.
>
> Reference:
> [1] Hanmeng Liu and Ruoxi Ning and Zhiyang Teng and Jian Liu and Qiji Zhou and Yue Zhang.2023.Evaluating the Logical Reasoning Ability of ChatGPT and GPT-4.arXiv

---

### Official Review · Reviewer_8wEx · 2023-08-05

**Soundness:** 4

**Excitement:**

4: Strong: This paper deepens the understanding of some phenomenon or lowers the barriers to an existing research direction.

**Paper Topic And Main Contributions:**

This paper proposes a scientific claim verification model based on a GNN model operating on causal structure. The proposed model constructs a tree representing causal structure for both claim and evidence, and compares the two structures to predict the label. The proposed model outperforms the current state of the art (T5) on HealthVER claim dataset.

**Questions For The Authors:**

The method uses SciBERT embedding layer - How important is this component in performance?

Are there any other graph-based baselines?

It would be interesting to see the effectiveness of the proposed model in low-label scenarios or domain generalization scenarios, compared to existing models -  Are such experiments possible?

**Reasons To Accept:**

The proposed model is well motivated and has good performance. Reasoning over causal structures in claim verification is an interesting research direction and the paper would encourage further research in this area.

The model is verified on two datasets, HealthVER and SciNLI.

The semantic-free models perform on par or better than PLM-based ones, and the model performs well with and without ground truth structure information.

**Reasons To Reject:**

My main concern is the limited number of evaluated datasets, and the clarity of writing.

**Reproducibility:**

4: Could mostly reproduce the results, but there may be some variation because of sample variance or minor variations in their interpretation of the protocol or method.

**Reviewer Confidence:**

3: Pretty sure, but there's a chance I missed something. Although I have a good feel for this area in general, I did not carefully check the paper's details, e.g., the math, experimental design, or novelty.

---

> ### Author Rebuttal · Authors · 2023-08-29
>
> We want to express our gratitude for your thorough review of our paper and for bringing forth significant questions. Your recognition of the uniqueness and importance of our approach is greatly valued. We provide answers to your questions below.
>
> **Q1**: The method uses SciBERT embedding layer - How important is this component in performance?
> **A1**: Thanks for raising this question. We are also interested in assessing the significance of SciBERT as an embedding layer. Hence, we have conducted the experiments to substitute SciBERT with BERT on $HealthVER_{truth}$ dataset. Here is the result:
> |          Model          |          F1(SciBERT)          | F1(BERT) |
> | ----- | ----- | ---- |
> | $StrOnly_{GCN}$ | 75.24 | 73.19 |
> | $StrOnly_{GAT}$ | 74.10 | 72.22 |
> | $VerQCS$ | 82.14 | 80.23 |
> From the results, we have the following observations:
> - The substitution leads to an average 2-point decrease in F1-score, indicating the crucial role of SciBERT in enhancing our model's performance.
> - VerQCS with SciBERT as the embedding layer outperforms PLMs using SciBERT. Even when replacing SciBERT with BERT as the embedding layer, VerQCS still outperforms PLMs using BERT, which demonstrates the effectiveness of VerQCS.
>
> **Q2**: Are there any other graph-based baselines?
> **A2**: We wish to kindly emphasize that VerQCS is the first work of entity-level graph-based approach in scientific claim verification. While we are aware of similar graph-based methods like GEAR[1], which uses GCNs with sentences as nodes, it's important to highlight that GEAR primarily focuses on paragraph-level evidence and doesn't delve into the causal structure within claims and evidence.
>
> **Q3**: It would be interesting to see the effectiveness of the proposed model in low-label scenarios or domain generalization scenarios, compared to existing models - Are such experiments possible?
> **A3**: We have devised a domain transfer experiment in which we assess the performance of models fine-tuned in HealthVER when applied directly to SciNLI in a zero-shot scenario. Here is the result:
>    | Model           | F1(fine-tuned) | F1(zero-shot) | Performance Drop | percent |
>    | --------------- | -------------- | ------------- | ---------------- | ------- |
>    | BERT            | 79.20          | 50.98         | 28.22            | 35.63   |
>    | SciBERT         | 80.13          | 53.19         | 26.94            | 33.62   |
>    | $StrOnly_{GCN}$ | 73.10          | 51.66         | 21.44            | 29.33   |
>    | $StrOnly_{GAT}$ | 70.89          | 50.42         | 20.47            | 28.88   |
>    | VerQCS          | 81.33          | 55.76         | 25.57            | 31.44   |
> The classification task differs between SciNLI and HealthVER. We combined entailment and reasoning into a support label, while the contrasting label is considered refuting, with neutral labels remaining unchanged. To ensure a fair comparison, we fine-tuned models in SciNLI using the same labels as HealthVER, resulting in an improved F1 score compared to the results in Table 2 of our paper. Our observations can be summarized as follows:
> - Adapting models fine-tuned on HealthVER for zero-shot evaluation in SciNLI resulted in an average F1-score drop of over 20 points. This experiment highlighted an interesting phenomenon: while semantic models showed greater performance degradation, models grounded in causal structure reasoning demonstrated greater robustness.
> - We find this difference reasonable since dataset distributions vary based on language style and construction. However, the difference in causal graphs is likely to be smaller than that in language.
>
> Once again, we extend our gratitude for your meticulous assessment of our paper.
>
> Reference:
> [1] Jie Zhou, Xu Han, Cheng Yang, Zhiyuan Liu, Lifeng Wang, Changcheng Li, and Maosong Sun. 2019.GEAR: Graph-based evidence aggregating and reasoning for fact verification. In Proceedings of the 57th Annual Meeting of the Association for Computational Linguistics, pages 892–901, Florence, Italy. Association for Computational Linguistics.

---

### Official Review · Reviewer_7skt · 2023-08-05

**Soundness:** 4

**Excitement:**

4: Strong: This paper deepens the understanding of some phenomenon or lowers the barriers to an existing research direction.

**Paper Topic And Main Contributions:**

The authors propose a novel pipeline for scientific claim verification leveraging qualitative causal structures “Verification by utilizing the Qualitative Causal Structure (VerQCS)”. Specifically, it extracts entities (which gets encoded by SciBERT for factor entity representations) and generates graphs for claims and evidence. Then, it calculates factor cross-attention for generating new claim and evidence graphs and decomposes them to tree structures. Then, those tree structures get homogenize to factor graphs and modifying graphs which get encoded as final tree embeddings. These embeddings get aggregated using cross-attention. Finally, the model predicts labels using both this structural representation and semantic features extracted by ‘claim [SEP] evidence’.

**Questions For The Authors:**

1. In Algorithm 1, where did you use flags? I see only setting flags to False but they are not used in anywhere.
2. What were the performances of proposed models for building the ground truth tree structures?


**Reasons To Accept:**

- The paper is well-written and the models are thoroughly analyzed by architectural variants and ablation tests.
- Proposed schema can capture more fine-grained
- The new model VerQCS obtained the highest performance for HEALTHVER dataset tasks.


**Reasons To Reject:**

- Baselines could be improved by fine-tuning or prompt tuning on HEALTHVER tasks for more fair comparison as newly proposed models were trained on these tasks specifically.

**Reproducibility:**

4: Could mostly reproduce the results, but there may be some variation because of sample variance or minor variations in their interpretation of the protocol or method.

**Reviewer Confidence:**

4: Quite sure. I tried to check the important points carefully. It's unlikely, though conceivable, that I missed something that should affect my ratings.

**Typos Grammar Style And Presentation Improvements:**

- P.2 L142-143 “COVID-19 ,according to (Sarrouti et al.,2021)” ->  “COVID-19. According to (Sarrouti et al.,2021)”
- L138 ‘2020),PubHealth’ -> ‘2020), PubHealth’
- L139 ‘2020),COVID-Fact’ -> L139 ‘2020),COVID-Fact’
- L140 ‘,HealthVer’ -> ‘ ,HealthVer’
- L143 ‘,according to’ -> ‘, according to’
- L183 ‘claims(‘ -> ‘, ‘claims (‘
- L216 ‘Factor remain’ -> ‘Factor remains’
- L235 ‘modifier,.' -> 'modifier.'
- L237 'comp_to,' -> 'comp_to'

---

> ### Author Rebuttal · Authors · 2023-08-29
>
> Thanks for your careful review of our paper and for raising important questions. We deeply appreciate your acknowledgment of the novelty and significance of our approach. Additionally, we are grateful for your patience and thoughtful feedback on typos, grammar, style, and presentation-related issues.  We provide answers to your questions below.
>
> **Q1**:Baselines could be improved by fine-tuning or prompt tuning on HEALTHVER tasks for more fair comparison as newly proposed models were trained on these tasks specifically.
> **A1**: With regard to the training of baselines, we would like to kindly clarify that they were indeed fine-tuned using the HealthVER training set, and the performance presented in the paper aligns comparably with that in the HealthVER paper. For further elaboration and comprehensive training specifics, we invite you to refer to the dataset section in Section 5.1 of our paper.
>
> **Q2**: In Algorithm 1, where did you use flags? I see only setting flags to False but they are not used in anywhere.
> **A2**: The notation "flags" in Algorithm 1 is utilized in cross-attention aggregation and serves to denote whether a tree, rooted by an association, qualifies as a root tree. Indeed, the "flag" notation is briefly mentioned in section 4.4 and we acknowledge that it lacks a comprehensive explanation within the algorithm. We've found a better way to rectify this and ensure that the role of the "flag" is clearly articulated. Your feedback has been invaluable in this regard, and we appreciate your attention to this matter.
>
> **Q3**: What were the performances of the proposed models for building the ground truth tree structures?
> **A3**: In consideration of the space constraints, we have included the performance results related to building the tree structures in Table 3, which can be found in Appendix A.1.
>
> Once again, we extend our gratitude for your meticulous assessment of our paper.

---

### Meta-Review · Area_Chair_E95x · 2023-09-18

**Recommendation:** 4

**Metareview:**

This paper proposes a scientific claim verification model based on a GNN model operating on causal structure. The proposed model constructs a tree representing causal structure for both claim and evidence, and compares the two structures to predict the label. The proposed model outperforms the current state of the art (T5) on HealthVER claim dataset.

The shared perspective of the reviewers include:
* the topic is very interesting
* the proposed model is well-motivated and the performance is extremely promising
* the work presented here is not only interesting within the scope of the paper, but it opens the door to research such as more fine-grained structures or in general the topic of reasoning over casual structures in claim verification

The issues of concern for all reviewers seem to be aspects of the paper or the analysis which can be addressed in a camera-ready, such as implementation using LLMs. In response, the authors seem to have thoroughly addressed each point clearly.

Based on the comments by the reviewers, I would hope the authors dedicate time to improve the overall clarity in the writing in a final version of the paper.

---

### Decision · Program_Chairs · 2023-10-07

**Decision:**

Accept-Main

**Comment:**

This paper proposes a scientific claim verification model based on a GNN model operating on causal structure. The proposed model constructs a tree representing causal structure for both claim and evidence, and compares the two structures to predict the label. The proposed model outperforms the current state of the art (T5) on HealthVER claim dataset.

The shared perspective of the reviewers include:
* the topic is very interesting
* the proposed model is well-motivated and the performance is extremely promising
* the work presented here is not only interesting within the scope of the paper, but it opens the door to research such as more fine-grained structures or in general the topic of reasoning over casual structures in claim verification

The issues of concern for all reviewers seem to be aspects of the paper or the analysis which can be addressed in a camera-ready, such as implementation using LLMs. In response, the authors seem to have thoroughly addressed each point clearly.

Based on the comments by the reviewers, I would hope the authors dedicate time to improve the overall clarity in the writing in a final version of the paper.